# Psychometric evaluation of the Asthma Diary Usability Questionnaire (ADU-Q): An adaptation and validation of a usability assessment tool among adults with Asthma

Sharifah Idayu Sayid Abdullah, Nik Munirah Nik Mohd Nasir ⓘ*, Lina Lohshini Kanoo ⓘ, Nur Amirah Shibraumalisi

Department of Primary Care Medicine, Faculty of Medicine, Universiti Teknologi MARA (UiTM), Sungai Buloh, Selangor, Malaysia

* nikmunirah@uitm.edu.my

## Abstract

The asthma diary has been established as a self-management tool to help patients achieve good disease control. However, asthma diary compliance is low, indicating a need to assess the diary's usability. Given the scarcity of a validated tool for assessing asthma diary usability, this study aims to adapt the Malay version of the mHealth App Usability Questionnaire (M-MAUQ) into the Asthma Diary Usability Questionnaire (ADU-Q) (Malay) and to assess its psychometric properties. In the first phase, the M-MAUQ items were adapted and rearranged into the four domains of the Nielsen usability model. In the second phase, content validity was assessed in two stages: domain validation by 3 experts, followed by item content validation by 7 experts. Face validation was then conducted by 10 patients with asthma. In the third phase, a cross-sectional study of 115 asthma patients attending the primary care clinic and the respiratory clinic follow-up was conducted. All patients completed the ADU-Q (Malay), and 62 patients were contacted 2 weeks later to complete the ADU-Q again. Exploratory factor analysis was performed with promax rotation, while reliability was assessed using Cronbach's alpha and the intraclass correlation coefficient. The item content validity index was 1.0, and item face validity indices ranged from 0.9 to 1.0. Exploratory factor analysis identified a three-factor structure with factor loadings >0.5 for all items; corrected item–total correlations exceeded 0.5, Cronbach's alpha was 0.982, and the intraclass correlation coefficient was 0.988, demonstrating excellent internal consistency and test–retest reliability. Based on these findings, ADU-Q is a valid, reliable and stable tool for assessing the usability of an asthma diary among adults with asthma.

which permits unrestricted use, distribution, and reproduction in any medium, provided the original author and source are credited.

**Data availability statement:** All relevant data are within the manuscript and its Supporting Information files.

**Funding:** The author(s) received no specific funding for this work.

**Competing interests:** The authors have declared that no competing interests exist.

## Introduction

Asthma, a chronic respiratory disease involving chronic airway inflammation, has been noted to gradually increase in prevalence since the 1990's [1]. 262 million people are affected by asthma worldwide [2], with the prevalence of asthma among the adult population in Asia varying between 2.5% to 7.5% [3]. In Malaysia, more than 1.4 million adults were reported to suffer from asthma, with 30% of them reported to have recurrent asthma attacks within 12 months period [4]. Another study found that more than 50% of adults with asthma had multiple emergency department visits in the past 12 months [5]. According to the RESONANCE study, which included individuals aged 12 years and above from a large populational databases in France, patients with uncontrolled asthma are two times more likely to be at risk of death compared to those with controlled asthma [6]. In terms of healthcare burden in Malaysia, approximately RM 301 (USD 68) was spent per patient for urgent care, compared with RM 177 (USD 40) for maintenance care [7].

To address the marked increase in prevalence and asthma deaths, asthma management guidelines were developed in the 1980's and 1990's and have been constantly updated until today. The guidelines highlighted the importance of patient empowerment and self-management, which include an asthma diary along with pharmacological treatment [8]. However, a study by Tsang et al demonstrated that compliance with completing the asthma diary was low, with only a quarter of participants still monitoring after six months [9].

Usability assessment of patient self-management tools is important to ensure that the tools are effective and efficient for patients to use [10]. According to the International Organization or Standardization (ISO), usability is defined as "the extent to which a product can be used to achieve specific goals with the effectiveness, efficiency, and satisfaction in a specified context of use" [11]. Usability assessment is important because it has been shown to help identify user issues, improve user-friendly interventions, enhance adherence, and reduce medical errors [12–14]. One of the established models of usability is the Nielsen Usability Model, which identifies five attributes of usability: learnability, efficiency, errors, satisfaction, and memorability [15]. Learnability was defined as the ease with which a novice user learns the system. Efficiency assesses how fast and easily a user performs a task after learning how to use it. The error domain assesses a product's ability to reduce user errors and the ease with which they can be corrected. The satisfaction domain assesses how pleasurable and comforting the product is for users. Memorability was defined as the ease with which each product function is remembered by the user [15].

Currently, there are questionnaires available to assess the usability of a tool; however, the items included in the questionnaire do not address the unique elements of the asthma self-management domain, which is predominantly based on the concept of patient-reported outcomes (PRO) [16]. Therefore, there is a need to develop a usability questionnaire that addresses specific elements of the asthma diary to help clinicians identify usability issues among asthma patients. This, in turn, will aid compliance and sustainability of this self-management tool.

For this purpose, the asthma diary used in this study was taken from the Asthma Self-Management Booklet©, originally developed in 2016 by Abdul Razak et al. (2016) and later updated in 2023 based on physicians' and patients' feedback. The booklet comprises patient education materials, clinic monitoring information, an asthma action plan and an asthma diary. To assess the usability of the asthma diary, this study aims to adapt and validate an asthma diary usability questionnaire for use among adults with asthma.

## Materials and methods

### Study design and population

This study was conducted in three phases. The first phase was divided into two stages: adapting the asthma diary usability questionnaire from the Malay-mHealth App Usability Questionnaire (M-MAUQ) and conducting content validation. The second phase was face validation, and the third phase was psychometric evaluation. The study population consisted of adult patients with asthma who attended a university-based primary care clinic and a respiratory clinic. This study has been approved by the Faculty Ethics Review Committee (FERC), Faculty of Medicine, UiTM, Reference: 100 – FPR (PT.9/19) (FERC-05-23-03).

### Phase 1: Adaptation and content validation

**Adaptation.** The questionnaire used in this adaptation is the Malay-language translation of the mHealth App Usability Questionnaire (M-MAUQ) [17]. The original English version was developed by Zhou et al. in 2019 to assess the usability of mobile health (mHealth) apps [18]. It comprises 18 items and is divided into 3 domains: "Ease of use" (5 items), "Interface and satisfaction" (7 items), and "Usefulness" (6 items). The M-MAUQ has a similar number of items to the original MAUQ but lacks allocated domains. The score is calculated using a 7-point Likert scale, which ranges from 1 (strongly disagree) to 7 (strongly agree). The total score is calculated by summing the item scores and then averaging them. The higher the average score, the better the usability; an average score below 4 indicates low usability. The Cronbach's Alpha for MAUQ was 0.914 [18], while the Malay-language translated MAUQ (M-MAUQ) also showed good reliability, with a Cronbach's Alpha> 0.90 [17]. Permission to adapt the M-MAUQ into the Asthma Diary Usability Questionnaire (ADU-Q) (Malay) was received from both the original MAUQ and M-MAUQ authors.

At this stage, all items in the original M-MAUQ were reviewed by the research team to identify terms and sentences specific to app usability. The usability element of the original item in M-MAUQ was identified through discussion among the research team prior to rewording or rephrasing. This was done to ensure the key usability element from the original questionnaire remained unchanged during the rewording or rephrasing process. Items with the term 'aplikasi' (app) were replaced with 'diari asma' (asthma diary). Items with sentences identified as general or specific for app use were rephrased to reflect use of the asthma diary. Finally, items irrelevant to the use of the asthma diary were removed from the questionnaire. This is consistent with the accepted adaptation process for questionnaires, whereby items that do not reflect the construct definition of the questionnaire can be deleted [19].

The adapted items, which were exclusively from the M-MAUQ, were then reallocated into four new domains based on the Nielsen usability model: Learnability, Efficiency, Satisfaction, and Error. The 'Memorability' domain was excluded because the diary contains no specific instructions requiring users to memorise before use. The original MAUQ did not specify a single usability model as the basis for the questionnaire. Nevertheless, a theoretical framework can help narrow the study's scope to specific variables [20]. The Nielsen Usability Model was used as a theoretical framework in this study because it is well established, developed by a prominent usability expert, and has served as a point of reference for subsequent usability models.

**Content validation.** Content validation was conducted in two stages. The first stage was domain validation, conducted via an online meeting with 3 experts: 1 family medicine specialist, 1 public health specialist, and 1 respiratory physician, each with more than 5 years of work experience. The family medicine specialist and respiratory physician have experience

managing patients at various stages of asthma, while the public health specialist has experience in public health and preventive medicine research on chronic diseases, including asthma. The experts were required to discuss and assess the relevance of the reallocated items under their respective new domains. The experts were provided with the definitions of the four usability domains, the original M-MAUQ, and the adapted ADU-Q questionnaire for review prior to the session. The assessment of the items in the domain validation was conducted through open-ended feedback. The research team made corrections to the adapted ADU-Q in response to experts' suggestions during the first round of domain validation. The revised, adapted ADU-Q underwent a second round of domain validation with the same experts via an online meeting to obtain feedback on the corrected domain allocation.

For the second stage, the adapted questionnaire underwent item content validation using a modified Delphi approach by 7 experts. The experts include 3 family medicine specialists, 1 respiratory physician, 1 public health specialist, 1 medical officer, and 1 pharmacist, all of whom have more than 5 years of experience working with patients with asthma through day-to-day clinic consultations, public health policies, and medication counselling and dispensing. The experts were emailed clear instructions for the study, a definition of the domain, a copy of the study tool, and the I-CVI score form. The experts were asked to return the I-CVI score form to the researcher via email within three days. The I-CVI form was checked for completion by the research team. The item was analyzed using the Item Content Validity Index (I-CVI), with experts scoring each item from 1 (not relevant) to 4 (very relevant) [19]. Items scoring 1 or 2 were re-categorized into "not relevant" with 0 points each. Items with a score of 3 or 4 were re-categorized into "relevant" with 1 point each. The average score of I-CVI for all items (S-CVI/Ave) was calculated to evaluate the relevance of the questionnaire. When there are three to five experts, the S-CVI/Ave should be 1.0 to have acceptable content validity [21]. The experts were also asked to provide suggestions for improving the questionnaire. Based on the calculated I-CVI and the expert panel's feedback, items deemed irrelevant or unclear were revised, modified, or excluded as appropriate. Items that scored 1 and 2 were omitted straight away as they were deemed not relevant. For items scored 3, the expert's opinion was sought before iteration. Once any modifications were made to the items, they were presented again to three experts from the original expert panel, who were randomly selected via simple random sampling, for another round of item content validation. The three experts were 1 respiratory physician and 2 family medicine specialists. The number of experts was reduced to three because no major corrections were required. The number of experts is acceptable, as the literature indicates that a minimum of two experts is required for content validation [21]. This process was discontinued after the second round, as the expert panel reached consensus on the final items of the ADU-Q. Reaching consensus is defined as "generally accepted opinion or decision among a group of people" [22]. For two experts, the acceptable I-CVI values were at least 0.80 [21]. In this case, all experts agree that the final items of the ADU-Q are relevant and reached an I-CVI of 0.8.

### Phase 2: Face validation

Face validity of the revised ADU-Q from Phase 1 was then assessed among patients with asthma attending the primary care clinic. A minimum of 10 samples is required to assess face validity [23]. Purposive sampling was used to select 10 participants who fulfilled the inclusion and exclusion criteria. The inclusion criteria were as follows: (a) diagnosed with bronchial asthma; (b) age more than 18 years old; (c) has been attending the primary care clinic or respiratory clinic for follow-up for at least 6 months; (d) has received and used the Asthma Self-Management Booklet© for at least 4 weeks; (e) able to read Malay language and (f) is Global initiative For Asthma (GINA) step 3 and below". The exclusion criteria include: (a) having any form of cognitive impairment or mental disorder which may impact one's capability in giving informed consent; (b) have physical disability which may impact the ability to write or read, and (c) diagnosed with bronchial asthma but with other concurrent lung diseases.

The participants were provided with a Google™ Form link to the ADU-Q. We asked participants to rate each item for clarity and understandability using a 1–4 scale (1 = not clear or understandable; 2 = somewhat clear or understandable; 3 = clear or understandable; 4 = very clear or understandable). The item-level face validity index (I-FVI) was computed for

each item by dichotomizing the 4-point scale. Items scoring 1 or 2 were re-categorised as "not clear and understandable" with 0 points. Items scoring 3 or 4 were re-categorized to "clear and understandable" with 1 point. The points (0 or 1) for each item were summed, and the total points were divided by the total number of respondents. An I-FVI value of at least 0.80 determined that the items were clear and well-understood [23]. For items scored 1 and 2, participants were encouraged to provide feedback on the item's lack of clarity. The items in ADU-Q were further refined before undergoing the psychometric evaluation.

### Phase 3: Field testing and psychometric evaluation

In this phase, the questionnaire's psychometric properties were evaluated through a cross-sectional study of patients with asthma attending the primary care and respiratory clinics. The inclusion and exclusion criteria were similar to those of Phase 2. However, participants who had previously participated in Phase 2 were excluded from Phase 3.

**Sample size determination.** The sample size for the final phase of ADU-Q was calculated based on the Sample to Variation ratio (SVR) of 6:1 [24]. The questionnaire comprises 16 items; the minimum required sample size was 96 participants. Considering a 20% non-response rate, the total sample size required was 115.

**Sampling method, patient recruitment and data collection.** Asthma patients were recruited via convenience sampling from the 2nd of January to the 30th of April 2024 at the primary care and respiratory clinics of a university hospital. The same researcher was assigned for data collection tasks throughout the period. Patients were approached on the day of their clinic appointment and were invited to participate in the study. A copy of the Asthma Self-Management Booklet© and an explanation of its use were provided to the patient. Patients who were interested in participating were provided with a study information sheet. The information sheet provides important details about the study, including the background, objectives, benefits, study protocol, confidentiality status, and contact information. For those who agreed to participate in the study, written consent was taken, and permission to contact the patient after 4 weeks to complete the questionnaire via Google™ form was obtained, along with the patient's contact number.

After 4 weeks of using the Asthma Self-Management Booklet©, the patient was contacted by phone and screened for inclusion and exclusion criteria through direct clarification to determine their eligibility for the study. Patients who met the eligibility criteria were sent a Google™ Form link via SMS to complete the ADU-Q self-administered. The participants were provided with clear verbal instructions for completing the questionnaire and were reminded to do so independently within 15 minutes. The participants were free to ask the researcher for further clarification. Upon completing the questionnaire, participants were asked to submit it to the researcher by clicking the "Send" button. The researcher checked the questionnaires for completion. If any responses were incomplete, participants were prompted to provide them upon consent.

**Data collection for test and retest.** The sample size for test-retest reliability is calculated using Bonet formula, giving a minimum sample size of 51 [25,26]. Considering a 20% non-response rate, the total sample size required for test-retest reliability is 62. Participants for test-retest reliability testing were selected via convenience sampling and contacted by phone 2 weeks after completing the questionnaires for the first time. The same link to the questionnaires was provided to participants to enable them to complete them.

### Statistical analysis

Data entry and statistical analysis were performed using IBM SPSS Statistics Version 29. The psychometric properties of ADU-Q were evaluated using External Factor Analysis and Reliability Analysis.

**Step 1 EFA: Assessment of data suitability.** The data were assessed for suitability prior to factor analysis. The suitability of the data was analysed via the Kaiser-Meyer-Olkin test (KMO), Bartlett's Test of Sphericity and anti-image matrix. The data was said to be adequate and suitable for analysis if the KMO value > 0.5, Bartlett's test p-value <0.05 and anti-image diagonal value >0.5 [27].

**Step 2: Factor extraction, retention and rotation.** In factor retention, many researchers use multiple criteria to decide on factor retention, which include parallel analysis, the Kaiser criterion (eigenvalue > 1), and the scree plot, with the latter two being the most commonly used [28]. In this study, factor retention was determined using parallel analysis, a scree plot, and a fixed number of factors. The Kaiser criterion method was not used as the original MAUQ has gone through psychometric analysis, which produced three factor solution [18]. Thus, the domain of items was preset to 3, consistent with the original questionnaire prior to factor analysis. Considering these two factors and the recommendation of the statistical experts, the fixed-factor method was chosen. By using dimensional reduction for factors in the extraction setting, the analysis of the data was set to a correlation matrix, the fixed numbers of factors to extract were set to the number of finalized domains in the first phase, and the display was set to an unrotated factor solution and a scree plot.

For the Monte Carlo parallel analysis, the data were analysed using SPSS and syntax. Random data was generated using syntax and compared with the real data on SPSS. After factor extraction and retention analysis, the data were rotated using promax rotation with the kappa value set to 4. Promax rotation was used because the domains were presumed to be related [29]. Due to the small sample size, the significance threshold for factor loadings was set at 0.6. Afthanorhan et al. highlighted the impact of sample size on factor loading and suggested using guidelines for significant factor loading based on sample size proposed by Hair et al. [30]. A factor loading > 0.7 indicates that the item's variance is sufficiently captured by the factor [31].

**Reliability analysis.** The reliability of ADU-Q was assessed using Cronbach's α, corrected item-total correlation (CITC), and intraclass correlation coefficient (ICC). For internal consistency, Cronbach's α and CITC were used to assess the ADU-Q. A minimum Cronbach's α coefficient value of 0.7 and CITC value of > 0.5 indicate that the questionnaire has good internal consistency [32]. ICC was used for test-retest reliability analysis. The ICC value was calculated using a two-way random-effects model with absolute agreement. An ICC value between 0.4 and 0.75 is considered as having good reliability, and above 0.75 as having excellent reliability [33].

The flow of the study is illustrated in Fig 1.

## Results

### Phase 1: Adaptation and content validation

**Adaptation.** In the adaptation stage, 11 items from M-MAUQ were reworded: the word "aplikasi" was replaced with "asthma diary" ("diari asma"), and 5 items underwent sentence rephrasing to address asthma diary usage. Item 8, which assesses feedback from an app, and item 11, which assesses app usage during internet disconnection [17], were not relevant when using an asthma diary booklet and were therefore removed, as both items were more relevant to app-based functionality rather than the hard-copy booklet. The total number of items after the adaptation stage was 16. The reallocation of items to the four domains resulted in 3 items in the Learnability domain, 4 in the Efficiency domain, 8 in the Satisfaction domain, and 1 in the Error domain.

**Content validation.** In the domain validation stage, all three experts agreed to move item 5 to the satisfaction domain. In contrast, two experts agreed that item 11 should be classified under the efficiency domain. Thus, both items were moved in accordance with the experts' recommendation. All three experts agreed to combine the error and learnability domains, as item 16 is the only item in the error domain. Thus, item 16 was moved under the learnability domain, and the domain was renamed to Learnability and Error. After two rounds of domain validation, 3 final domains remained, and the I-CVI was 1 for all items in the ADU-Q.

In the item validation stage, 15 out of 16 items achieved an I-CVI of 1. Items 4 and 5 achieved an I-CVI of 1, but two experts suggested adding examples to clarify them. One expert suggested adding an alternative word for access, "akses", to the sentence in Item 8. The experts noted that all items are relevant for assessing the usability of an asthma diary. Following the corrections, the items underwent a second round of item validation. After the second round of item validation,

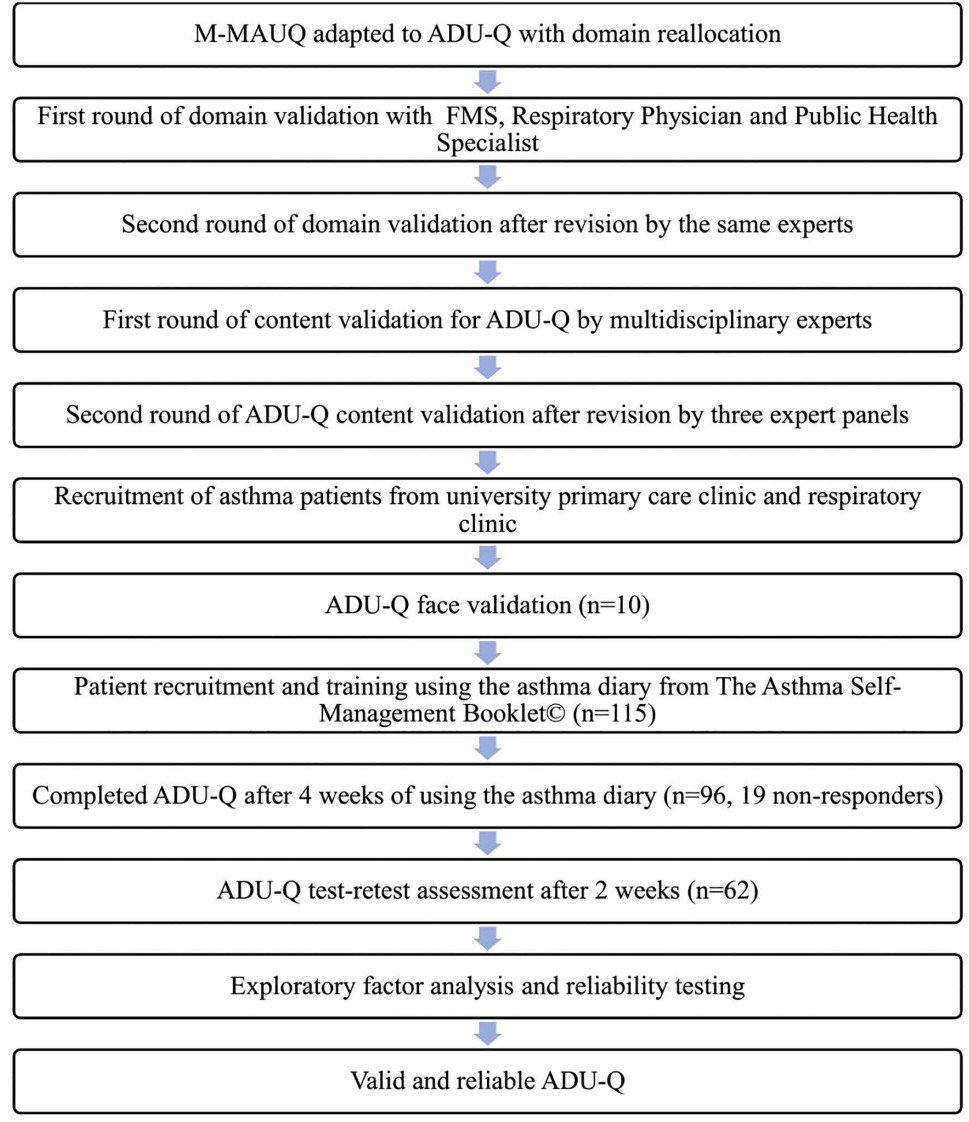

**Fig 1. Study flowchart.** M-MAUQ: mHealth App Usability/questionnaire (Malay version). ADU-Q: Asthma Diary Usability Questionnaire. FMS: Family Medicine Specialist.

the experts retained all 16 items for ADU-Q, with the calculated S-CVI/Ave reaching 1.0. Table 1 presents the I-CVI scores for the second-stage content validation of the ADU-Q.

## Phase 2: Face validation of ADU-Q

During face validation, the majority of the participants scored the items between 3 and 4. Five participants stated that the items were clear and simple to answer. Only one participant scored 2 on items 14 and 15, but commented that all items were clear and easy to understand. The calculated I-FVI for both questionnaires ranged between 0.9 and 1.0 for each item. Thus, no amendments were made, and ADU-Q proceeded to psychometric evaluation in Phase 3. Table 2 show-cases the results of I-FVI for ADU-Q.

**Table 1. Final round content validity index (second stage) for item validation for ADU-Q.**

| Item | Experts | | | No. of experts in agreement | I-CVI |
|---|---|---|---|---|---|
| | E1 | E2 | E3 | | |
| LE 1 | 4 | 4 | 4 | 3 | 1.0 |
| LE 2 | 4 | 4 | 4 | 3 | 1.0 |
| LE 3 | 4 | 4 | 4 | 3 | 1.0 |
| LE 4 | 4 | 3 | 4 | 3 | 1.0 |
| EF 5 | 4 | 4 | 4 | 3 | 1.0 |
| EF 6 | 4 | 4 | 4 | 3 | 1.0 |
| EF 7 | 4 | 4 | 4 | 3 | 1.0 |
| EF 8 | 4 | 4 | 4 | 3 | 1.0 |
| EF 9 | 4 | 4 | 4 | 3 | 1.0 |
| SAT 10 | 4 | 4 | 4 | 3 | 1.0 |
| SAT 11 | 4 | 4 | 4 | 3 | 1.0 |
| SAT 12 | 4 | 3 | 4 | 3 | 1.0 |
| SAT 13 | 4 | 3 | 4 | 3 | 1.0 |
| SAT 14 | 4 | 4 | 4 | 3 | 1.0 |
| SAT 15 | 4 | 4 | 4 | 3 | 1.0 |
| SAT 16 | 4 | 4 | 4 | 3 | 1.0 |
| S-CVI/Ave | | | | | 1.0 |

I-CVI = 1 indicate good content validity.

E = Experts I-CVI = Item content validity index.

LE = Learnability and error domain, EF = Efficiency domain, SAT = Satisfaction domain.

## Phase 3: Psychometric Analysis of ADU-Q

A total of 115 patients were recruited and provided consent to participate in the study. After 4 weeks of using the Asthma Self-Management Booklet©, only 96 participants completed the ADU-Q. The other 19 participants could not be contacted during the study period despite repeated attempts and were considered non-responders, accounting for a 16.5% non-response rate.

**Demographic and clinical characteristics of participants.** The participants' ages ranged from 23 to 82 years old. The mean age was 52.6 (SD ± 14.9). Most participants were Malay (97.9%) and female (78.1%). Regarding asthma, most participants were under respiratory clinic follow-up (55.2%), had asthma for more than 10 years (66.7%), were at GINA step 3 (83.3%), and used a formoterol fumarate plus budesonide inhaler (39.6%). The average asthma control test (ACT) score seen in this study was 21.2 (±3.38). The demographic and clinical characteristics of participants in this study are summarized in Table 3.

**Psychometric properties.** The psychometric evaluation included 96 responses. All responses were complete, with no missing responses, indicating good data quality. The KMO value was 0.942; Bartlett's test of sphericity ($p < 0.01$); and all anti-image diagonal values were >0.5, indicating that the data were suitable for analysis. The highest communalities were for item 2, with a value of 0.968, indicating that it accounts for 96% of its variability.

All items in each questionnaire underwent Principal Component Analysis, followed by Promax rotations. The scree plot shows an elbow at the second, third, and fourth components, indicating the retention of three factors. In the parallel Monte Carlo analysis, the Eigenvalues of only one factor exceed both the mean and the 95th-percentile Eigenvalues. In light of the original MAUQ having 3 domains and supported by the scree plot findings, the research team decided to retain 3 factors.

**Table 2. Face validity index for ADU-Q.**

| Item | Raters | | | | | | | | | | Raters in Agreement | I-FVI |
|---|---|---|---|---|---|---|---|---|---|---|---|---|
| | R1 | R2 | R3 | R4 | R5 | R6 | R7 | R8 | R9 | R10 | | |
| LE 1 | 4 | 4 | 3 | 3 | 4 | 4 | 4 | 4 | 3 | 3 | 10 | 1.0 |
| LE 2 | 4 | 4 | 3 | 3 | 4 | 4 | 4 | 4 | 3 | 3 | 10 | 1.0 |
| LE 3 | 4 | 4 | 3 | 3 | 4 | 4 | 4 | 4 | 3 | 3 | 10 | 1.0 |
| LE 4 | 4 | 4 | 4 | 3 | 4 | 4 | 4 | 4 | 3 | 3 | 10 | 1.0 |
| EF 5 | 4 | 4 | 4 | 3 | 4 | 4 | 4 | 4 | 3 | 3 | 10 | 1.0 |
| EF 6 | 4 | 4 | 4 | 3 | 4 | 4 | 4 | 4 | 3 | 3 | 10 | 1.0 |
| EF 7 | 4 | 4 | 3 | 3 | 4 | 4 | 4 | 4 | 3 | 3 | 10 | 1.0 |
| EF 8 | 4 | 4 | 3 | 3 | 4 | 4 | 4 | 4 | 3 | 3 | 10 | 1.0 |
| EF 9 | 4 | 4 | 3 | 3 | 4 | 4 | 4 | 4 | 3 | 3 | 10 | 1.0 |
| SAT 10 | 4 | 4 | 3 | 3 | 4 | 4 | 4 | 4 | 3 | 3 | 10 | 1.0 |
| SAT 11 | 4 | 4 | 3 | 3 | 4 | 4 | 4 | 4 | 3 | 3 | 10 | 1.0 |
| SAT 12 | 4 | 4 | 3 | 3 | 4 | 4 | 4 | 4 | 3 | 3 | 10 | 1.0 |
| SAT 13 | 4 | 4 | 3 | 3 | 4 | 4 | 4 | 4 | 3 | 3 | 10 | 1.0 |
| SAT 14 | 4 | 4 | 4 | 3 | 4 | 4 | 4 | 4 | 2 | 3 | 9 | 0.9 |
| SAT 15 | 4 | 4 | 3 | 3 | 4 | 4 | 4 | 4 | 2 | 3 | 9 | 0.9 |
| SAT 16 | 4 | 4 | 4 | 3 | 4 | 4 | 4 | 4 | 3 | 3 | 10 | 1.0 |

I-FVI ≥ 0.8 indicates good face validity.

R = rater I-FVI = Item face validity index.

LE = Learnability and error domain, EF = Efficiency domain, SAT = Satisfaction domain.

For factor loading, item loadings ranged from 0.63 to 0.939, indicating strong correlations with the factor. Factor 1, with the highest cumulative variance of 79% were loaded with items 10–16. Factor 2, with the second-highest cumulative variance (7.7%), was loaded with items 1–4. Factor 3, with a cumulative variance of 5.2% were loaded with items 5–9. All items had factor loadings> 0.6, indicating that they are valid and can be used to measure the usability of an asthma diary. Table 4 shows the loading value results for ADU-Q.

**Reliability analysis.** Reliability was assessed using corrected item-total correlation (CITC) and Cronbach's Alpha. The calculated CITC ranged from 0.813 to 0.911, and the overall Cronbach's Alpha was 0.982. The Cronbach Alpha for Factor 1 was 0.980, Factor 2 was 0.983 and Factor 3 was 0.968. Table 5 shows the result of the analysis.

For test-retest reliability, the overall ICC value for ADU-Q was 0.988, indicating good reproducibility. Individual items had ICC values ranging from 0.725 to 0.985. Table 6 presents the ICC values for each item.

## Discussion

To the best of our knowledge, this is the first validation study of a questionnaire assessing the usability of a written asthma diary among patients with asthma attending primary care and respiratory clinics in Malaysia.

The participants in this study include patients attending either a primary care clinic or a respiratory clinic. Although most asthma patients receive care in primary care clinics rather than secondary care facilities [34,35], this study also includes patients from the respiratory clinic, as the use of an asthma diary is universal across health care settings. However, to ensure homogeneity, the inclusion criteria were set at GINA step 3 or below, and participants with other underlying lung diseases were excluded. The mean age of the participants was 52.6 (±14.9), which was higher than the mean age of asthma patients in the country [5,36]. The Asthma Control Test (ACT) score among participants in this study was 21.2, indicating well-controlled asthma, consistent with other asthma studies in the country [37].

**Table 3. Demographics and clinical characteristics of participants (n = 96).**

| Characteristics of the participants | Frequency, n (%) | Mean (±SD) |
|---|---|---|
| **Age (years)** | | 52.6 (±14.9) |
| **Gender** | | |
| Male | 21 (21.8) | |
| Female | 75 (78.1) | |
| **Ethnicity** | | |
| Malay | 94 (97.9) | |
| Others | 2 (2.1) | |
| **Clinic type** | | |
| Respiratory clinic | 53 (55.2) | |
| Primary care clinic | 43 (44.8) | |
| **Asthma duration** | | |
| Less than 5 years | 17 (17.7) | |
| 5 to 10 years | 15 (15.6) | |
| More than 10 years | 64 (66.7) | |
| **GINA** | | |
| Step 1 | 7 (7.3) | |
| Step 2 | 9 (9.4) | |
| Step 3 | 80 (83.3) | |
| **Asthma control test (ACT) score** | | 21.2 (±3.38) |
| **Asthma inhalers** | | |
| Salbutamol and budesonide | 11 (11.5) | |
| Salbutamol and Fluticasone | 1 (1) | |
| Salmeterol + fluticasone propionate and Salbutamol | 18 (18.8) | |
| Formoterol fumarate + budesonide | 38 (39.6) | |
| Formoterol fumarate + beclomethasone dipropionate | 28 (29.2) | |

The original MAUQ had 18 items, distributed across 3 domains: 'Ease of use', 'Interface and satisfaction', and 'Usefulness'. The translated Malay-language version, M-MAUQ, similarly had 18 items but no allocated domains [17,18]. During the adaptation of M-MAUQ to ADU-Q, 2 irrelevant items were removed. A similar approach has been conducted by other questionnaire adaptation studies, where items have been removed as it does not reflect the construct of the adapted questionnaire [38,39]. Nevertheless, the interpretation of the usability score was unaffected, as the interpretation for M-MAUQ is based on the average score.

The remaining 16 items were initially mapped to four domains based on the Nielsen Usability Model components: Learnability (3 items), Efficiency (4 items, Satisfaction (8 items) and Error (1 item). After review by the expert panels during content validation, the sole item under the Error domain was moved to the Learnability domain and renamed to Learnability and Error (4 items). This aligns with recommendations in the literature to include at least 3 items per factor for all subscales to be successfully identified [40]. The final 3 remaining domains and corresponding items are deemed relevant and important by the expert panels in assessing the usability of an asthma diary. Subsequent face validation by the participants also concluded that all items were clear and easy to understand. This information is important because patients may interpret items differently from how the questionnaire was intended, leading to response error [41].

The final validated ADU-Q (S1 File. Final ADU-Q items) exhibits good internal consistency, as indicated by corrected total-item correlations (CITC) ranging from 0.813 to 0.913 and an overall Cronbach's Alpha of 0.982. This is comparable

**Table 4. Factor loading values ADU-Q.**

| Item | Question | Factor 1 | Factor 2 | Factor 3 |
|------|----------|----------|----------|----------|
| LE 1 | It was easy for me to learn to use the asthma diary. | | 0.910 | |
| LE 2 | The asthma diary was easy to use | | 0.939 | |
| LE 3 | The format of the asthma diary was easy to follow | | 0.928 | |
| LE 4 | Whenever I made a mistake using the asthma diary, I could correct it easily | | 0.909 | |
| EF 5 | The asthma diary format allowed me to record my asthma symptoms easily | | | 0.658 |
| EF 6 | The information in the asthma diary was well organized, so I could easily find the information I needed | | | 0.827 |
| EF 7 | The amount of time involved in using the asthma diary has been fitting for me | | | 0.770 |
| EF 8 | The asthma diary improves my access/reach to health care services (Example: providing the record of asthma symptom to the treating doctor) | | | 0.857 |
| EF 9 | The asthma diary helped me manage my asthma effectively | | | 0.887 |
| SAT 10 | I like the format of the asthma diary | 0.630 | | |
| SAT 11 | The asthma diary would be useful for my health and well-being | 0.826 | | |
| SAT 12 | This asthma diary has all the functions and capabilities I expected it to have (Example: recording asthma symptoms at home) | 0.905 | | |
| SAT 13 | This asthma diary provided an acceptable way to receive health care services, such as performing self-assessment. | 0.929 | | |
| SAT 14 | I feel comfortable using this asthma diary in social settings, even with the presence of friends and family | 0.914 | | |
| SAT 15 | I would use this asthma diary again | 0.877 | | |
| SAT 16 | Overall, I am satisfied with this asthma diary | 0.869 | | |

LE = Learnability and error domain, EF = Efficiency domain, SAT = Satisfaction domain

to the M-MAUQ, which was tested among 51 university students in the International Islamic University of Malaysia, which had the overall Cronbach's Alpha of 0.946 [17] and the English standalone MAUQ, which was tested among 128 participants in the US, which had the overall Cronbach's Alpha of 0.914 [18]. In terms of test-retest reliability, individual items' ICC values ranged between 0.725 and 0.985 with an overall ICC value of >0.9, which is higher compared to a German adaptation and translation of MAUQ in another study, which is 0.75 [42].

## Strength and limitation

This study's strengths include an 83.4% response rate and high-quality data, with no missing responses and no cross-loading items. This questionnaire also addresses a gap in the literature by evaluating the usability of an asthma diary assessment tool, particularly in Malay. Thus, it is suitable for assessing the usability of a locally developed asthma diary in Malaysia.

There are limitations to this study. First, the results of this study should be interpreted with caution, as the majority of participants were female (78.1%). Another limitation of this study is its use of convenience sampling, which may introduce sampling bias. Third, only individuals who can read and understand Malay can use the questionnaires. To evaluate the effect of this on ADU-Q, a confirmatory factor analysis is recommended. Reliance on patient self-reporting represents a limitation of this study; however, as usability is inherently based on user self-reflection, self-report measures were considered the most appropriate assessment method.

**Table 5. Total-item statistics.**

| | Scale Mean if Item deleted | Scale Variance if Item Deleted | Corrected Item-Total Correlation | Squared Multiple Correlation | Cronbach's Alpha if Item Deleted |
|---|---|---|---|---|---|
| LE 1 | 83.26 | 326.868 | 0.858 | 0.961 | 0.981 |
| LE 2 | 83.27 | 326.873 | 0.857 | 0.969 | 0.981 |
| LE 3 | 83.33 | 327.446 | 0.855 | 0.944 | 0.981 |
| LE 4 | 83.32 | 326.958 | 0.829 | 0.896 | 0.981 |
| EF 5 | 83.39 | 328.218 | 0.813 | 0.839 | 0.981 |
| EF 6 | 83.48 | 322.105 | 0.879 | 0.921 | 0.980 |
| EF 7 | 83.59 | 316.665 | 0.904 | 0.944 | 0.980 |
| EF 8 | 83.46 | 319.367 | 0.895 | 0.947 | 0.980 |
| EF 9 | 83.51 | 318.231 | 0.835 | 0.889 | 0.981 |
| SAT 10 | 83.50 | 325.916 | 0.876 | 0.886 | 0.980 |
| SAT 11 | 83.48 | 321.705 | 0.850 | 0.860 | 0.981 |
| SAT 12 | 83.48 | 319.873 | 0.911 | 0.957 | 0.980 |
| SAT 13 | 83.42 | 319.677 | 0.906 | 0.958 | 0.980 |
| SAT 14 | 83.59 | 320.054 | 0.858 | 0.880 | 0.981 |
| SAT 15 | 83.50 | 317.389 | 0.911 | 0.952 | 0.980 |
| SAT 16 | 83.42 | 317.004 | 0.901 | 0.944 | 0.980 |
| **Overall Cronbach Alpha** | | | | | 0.982 |

LE = Learnability and error domain, EF = Efficiency domain, SAT = Satisfaction domain

**Table 6. ICC value ADU-Q.**

| Item | ICC | Range |
|---|---|---|
| LE 1 | 0.958 | 0.926–0.976 |
| LE 2 | 0.725 | 0.519–0.843 |
| LE 3 | 0.962 | 0.934–0.979 |
| LE 4 | 0.966 | 0.941–0.981 |
| EF 5 | 0.881 | 0.792–0.932 |
| EF 6 | 0.972 | 0.951–0.984 |
| EF 7 | 0.974 | 0.955–0.985 |
| EF 8 | 0.977 | 0.960–0.987 |
| EF 9 | 0.975 | 0.956–0.986 |
| SAT 10 | 0.985 | 0.973–0.991 |
| SAT 11 | 0.979 | 0.963–0.988 |
| SAT 12 | 0.977 | 0.959–0.987 |
| SAT 13 | 0.971 | 0.950–0.984 |
| SAT 14 | 0.947 | 0.907–0.970 |
| SAT 15 | 0.941 | 0.958–0.986 |
| SAT 16 | 0.964 | 0.936-0.979 |

LE = Learnability and error domain, EF = Efficiency domain, SAT = Satisfaction domain

## Conclusion

ADU-Q is a valid and reliable instrument for assessing the usability of an asthma diary. The questionnaire comprises 16 items and demonstrates good construct validity, high internal consistency, and excellent test-retest reliability. The questionnaire can be further used to assess the usability of an asthma diary from the perspectives of learnability, efficiency, and satisfaction.

## Supporting information

**S1 File. Final ADU-Q items.**
(DOCX)

## Acknowledgments

The authors wish to thank Mr Zhou L, the author of the original MAUQ, and Dr Nik Shanita, the author of M-MAUQ, for allowing the questionnaires to be adapted for the ADU-Q.

## Author contributions

**Conceptualization:** Nik Munirah Nik Mohd Nasir, Lina Lohshini Kanoo.

**Data curation:** Sharifah Idayu Sayid Abdullah.

**Formal analysis:** Sharifah Idayu Sayid Abdullah.

**Investigation:** Sharifah Idayu Sayid Abdullah.

**Methodology:** Sharifah Idayu Sayid Abdullah, Nik Munirah Nik Mohd Nasir, Lina Lohshini Kanoo, Nur Amirah Shibraumalisi.

**Resources:** Nik Munirah Nik Mohd Nasir.

**Supervision:** Nik Munirah Nik Mohd Nasir, Lina Lohshini Kanoo.

**Writing – original draft:** Sharifah Idayu Sayid Abdullah.

**Writing – review & editing:** Sharifah Idayu Sayid Abdullah, Nik Munirah Nik Mohd Nasir, Lina Lohshini Kanoo, Nur Amirah Shibraumalisi.

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
