## [Decision Letter · Decision Letter 0]

5 Nov 2024

Dear Dr. Nik Mohd Nasir,

We look forward to receiving your revised manuscript.

Kind regards,

Shairyzah Ahmad Hisham, PhD.

Academic Editor

PLOS ONE

Journal Requirements: When submitting your revision, we need you to address these additional requirements. 1. Please ensure that your manuscript meets PLOS ONE's style requirements, including those for file naming. The PLOS ONE style templates can be found at https://journals.plos.org/plosone/s/file?id=wjVg/PLOSOne_formatting_sample_main_body.pdf and https://journals.plos.org/plosone/s/file?id=ba62/PLOSOne_formatting_sample_title_authors_affiliations.pdf 2. PLOS ONE has specific requirements for studies that are presenting a new method or tool as the primary focus, including a newly developed or modified questionnaire or scale (https://journals.plos.org/plosone/s/submission-guidelines#loc-methods-software-databases-and-tools.) One requirement is that the questionnaire or scale must be openly available under a license no more restrictive than CC BY. In light of this, before we proceed, please include a copy of your questionnaire or scale as a Supporting Information file or provide a link if it is available through an online repository. By uploading a copy of the questionnaire you have used for this study, you are confirming that you have obtained all the necessary permissions to publish the questionnaire CC BY and have included all proper attributions in your manuscript. For more detail, please see here: https://journals.plos.org/plosone/s/licenses-and-copyright. If you are unable to obtain the necessary permissions, please contact us so that we can withdraw your manuscript from consideration. 3. Your ethics statement should only appear in the Methods section of your manuscript. If your ethics statement is written in any section besides the Methods, please move it to the Methods section and delete it from any other section. Please ensure that your ethics statement is included in your manuscript, as the ethics statement entered into the online submission form will not be published alongside your manuscript. 4. Please include captions for your Supporting Information files at the end of your manuscript, and update any in-text citations to match accordingly. Please see our Supporting Information guidelines for more information: http://journals.plos.org/plosone/s/supporting-information.

Reviewers' comments:

Reviewer's Responses to Questions

**Comments to the Author**

1. Is the manuscript technically sound, and do the data support the conclusions?

Reviewer #1: Yes

Reviewer #2: Yes

2. Has the statistical analysis been performed appropriately and rigorously?

Reviewer #1: Yes

Reviewer #2: No

3. Have the authors made all data underlying the findings in their manuscript fully available?

Reviewer #1: Yes

Reviewer #2: Yes

4. Is the manuscript presented in an intelligible fashion and written in standard English?

Reviewer #1: Yes

Reviewer #2: Yes

Reviewer #1: Manuscript Review for PONE-D-24-32929

Dear authors,

Congratulations on your research and on producing the manuscript. My comments serve to improve the clarity of reporting and ease comprehension for new readers.

Title:

The title is clear and reflects the objective and methods of your research.

Abstract:

The abstract was clear, structured and comprehensible.

Introduction:

Introduction was well-funneled and well-written in highlighting the background and why such research is needed. The main objective was clearly stated. No additional improvements necessary.

Methods:

1. Figure 1 contains a lot of abbreviation. Please include footnotes or relocate to the end of the Methods section as a section summary.

Section 2.2 Study tool

1. Please state the language used in ADU-Q.

Section 2.2.3

Assessment of domain validity

1. Please explain how assessment of domain validity was conducted; i.e. rating, open-ended feedback, etc.

2. Please state, in general, the years of working experience of the 3 domain validity assessors and whether they had extensive experience with asthma patients/management.

3. Please state whether any iterations to the ADU-Q after domain validity assessments were made after the first round.

Assessment of item validity

1. Please state, in general, the years of working experience of the item validity assessors and whether they had extensive experience with asthma patients/management.

2. Please clarify whether the study method of assessing content validity is a form a modified-Delphi or otherwise. If yes, please indicate in the manuscript.

3. Please clarify and cite the cutoff S-CVI/Ave value to indicate content validity.

4. Please indicate and cite the definition of “reaching consensus”.

5. In the flow chart, the second round of content validation was only conducted by 3 out of the 7 panels. Please restate this in the paragraph: who were the three and whether the reduction of the number of panels were permissible/good practice.

2.4 Face validation

1. Please omit the word asthma control and rewrite “(f) having the level of asthma control at Global initiative For Asthma (GINA) step 3 and below” to “(f) is Global initiative For Asthma (GINA) step 3 and below”. This is because “asthma control” refers to controlled, partially controlled and uncontrolled whereas the GINA steps refer to “asthma management”.

2.6.2. Factor extraction

1. Please elaborate and cite the decision to use the fixed number of factors method.

Results

1. Table 1: Final round content validity index (second stage) for item validation for ADU-Q Please clarify the number of experts who participated in the final round of the content validation; in figure 1 study flowchart reported 3 experts but table 1 presented results from all 7 expert.

2. Table 3 – Please substitute the word (GINA) “Stage” to “Step” in this table and throughout the manuscript.

Discussion

1. Page 25, first paragraph: “The mean average usability score for ADU-Q was 5.58 (SD ±1.19)”. Please clarify where readers could find this in the results section. Please elaborate on the total score permissible for the ADU-Q and what scores mean (higher, lower, categories, if any)

2. Page 25, first paragraph: “The mean average usability score for ADU-Q was 5.58 (SD ±1.19)”. Please clarify whether this reflects the usability of the ‘Buku Urus Kendiri Asma’ or something else.

3. Page 27, first paragraph: “In factor retention, several methods are used…”. Please consider moving this paragraph to a relevant subsection under Methods section.

Limitations

1. “There are four limitations to this study.” Please check and make clear whether you have listed these four limitations.

2. Please clarify on whether or not the small number of expert panels in the content validation stage is a potential limitation.

General comments:

Methods, Results and Discussions can be re-written more concisely. I recommend touching-up the manuscript for typos i.e. “dairy (page 3)”, grammatical errors and better sentence structure, for the readers’ benefits.

Congratulations again on this meaningful research and manuscript. Thank you.

Reviewer #2: Dear Authors,

Overall the research is relevant as currently there are hardly any studies being conducted to assess the usability of an asthma self-management tool. The following are the comments according to sub sections:

Abstract

The abstract should elaborate more on the background of the study. Perhaps including the current situation on the control of asthma among the Malaysian population which further emphasize the urge to conduct this research. The content of abstract was not equally divided according to subsection. Most of the content are elaborating on the methods part. Perhaps the results and discussion parts should be refine in the abstract for the readers to get a better overview of the research.

Introduction

In the first paragraph of the introduction, is it possible to add more details (RESONANCE study, no. of patients included with age range and other details such as comorbidities). The details need not be too long, it can be summarised within one or two sentences. In the paragraph, it was also mentioned ' In terms of healthcare burden, the cost to treat patients with uncontrolled asthma is higher than those with controlled asthma'. Is this sentence referring to the Malaysian population or other studies?. The following sentence on the costs of urgent care and maintenance care in Malaysia is also confusing as the costs were stated in USD. Perhaps the costs in both currencies can be stated.

Paragraph 3 and 4 are disconnected. The use of asthma self-management booklet was developed in 2016 and there had been an updated version in 2023 indicating some precipitating issues/problems leading to the update of the booklet. This could be related to the need for usability assessment. Elaboration on this will make a better flow.

Methods

I found that this part of the manuscript contains sufficient details of the conduct of the study. In figure 1, FMS was first mentioned. The author may want to spell out FMS in figure 1 and use abbreviations in the following paragraphs.

Results

Results were appropriately presented. However, i have preference in terms of presentation of table 4, table 5, table 6 e.g. putting the reference range underneath each table for ease of reference (factor loading>0.6 indicates items validity). I noticed there are a lot of results presented but not discussed in the discussion i.e. description/elaboration on the population distribution male vs female, reasoning on why there are more malay population etc. Another point for discussion would be conducting research in a semi-government university hospital would lead to different treatment approach hence affecting the GINA stages and treatment options as well as ACT score. the population's asthma duration can also be discussed further.

Discussion

Under discussion, there were several definitions given i.e questionnaire, good questionnaire. It could have been mentioned in the introduction instead to allow other important points to be included in the discussion e.g. elaboration on the results. Some parts of the discussion are redundant with the results for example "Test with Cronbach Alpha of >0.7 is considered reliable" and "overall ICC value was > 0.9 indicating good reproducibility and stability of the

score over time.

Strength and Limitation

In the results part it was stated that during the data collection, a copy of the Asthma Self-Management Booklet and an explanation regarding the usage of the book were given to the patient. Would there be any bias imposed to the data collection process.

Statistical Analysis

Psychometric properties - It was stated that all items in the questionnaire went through principal component analysis. Is this related to determining whether the factors are correlated before deciding to use promax rotations for analysis?. Is learnability strongly correlated with efficiency and satisfaction? etc.

That is the end of my comments. It is hoped that the issues can be addressed to enhance the readers' interest. Thank you.

**Do you want your identity to be public for this peer review?** For information about this choice, including consent withdrawal, please see our Privacy Policy

Reviewer #1: No

Reviewer #2: **Yes:** Aina Yazrin Ali Nasiruddin

---

## [Author Response · Author response to Decision Letter 1]

28 Jul 2025

Response to reviewers

REVIEWER 1

Comment

Figure 1 contains a lot of abbreviation. Please include footnotes or relocate to the end of the Methods section as a section summary.

Figure 1 has been shifted to the end of method section and the footnotes for the abbreviation has been added (page 15)

Please state the language used in ADU-Q

The language used in ADU-Q is Malay language. This has been added to line 1, page 6.

“Permission to adapt the M-MAUQ into the Asthma Diary Usability Questionnaire (ADU-Q) (Malay) was received from both the original MAUQ and M-MAUQ authors. “

Domain validity

Please explain how assessment of domain validity was conducted; i.e. rating, open-ended feedback, etc.

Please state whether any iterations to the ADU-Q after domain validity assessments were made after the first round.

The conduct of domain validity assessment has been added under Content validation section (page 7)

“The assessment of the items in domain validation was done via open ended feedback.”

Yes this has been stated under Content validation section (page 7-8)

“Corrections to the adapted ADU-Q were made by the research team based on the suggestion from the experts during first round of domain validation.”

Please state, in general, the years of working experience of the 3 domain validity assessors and whether they had extensive experience with asthma patients/management.

Years of experience including their experience with asthma patients/management has been added under Content validation section (page 7)

“…3 experts consisting of 1 family medicine specialist, 1 public health specialist and 1 respiratory physician with working experience of more than 5 years. The family medicine specialist and respiratory physician have experience in managing patients with various stages of asthma while the public health specialist has experience in public health and preventive medicine research for chronic diseases which includes asthma.”

Item validity

Please clarify whether the study method of assessing content validity is a form a modified-Delphi or otherwise. If yes, please indicate in the manuscript

The study method for content validity is based on modified-Delphi. This has been added in the paragraph on page 8.

“For the second stage, the adapted questionnaire went through item content validation process using modified-Delphi approach by 7 experts.”

Please clarify and cite the cutoff S-CVI/Ave value to indicate content validity

The cutoff point and citation for the S-CVI/Ave has been added to the Content validation section, page 8.

“When there are three to five experts, the S-CVI/Ave should be 1.0 to have acceptable content validity (29).”

Please indicate and cite the definition of “reaching consensus”

The definition and citation for “reaching consensus” added to the Content validation section, page 9.

“Reaching consensus is defined as “generally accepted opinion or decision among a group of people” (30). In this case, all experts agree that the final items of the ADU-Q is relevant.

In the flow chart, the second round of content validation was only conducted by 3 out of the 7 panels. Please restate this in the paragraph: who were the three and whether the reduction of the number of panels were permissible/good practice.

This explanation has been added under the Content validation section, page 8-9.

“The number of experts was reduced to three as there were no major corrections required. This number of experts are acceptable, as based on literature, the minimum number of experts for content validation are at least two (29).”

Face validation

Please omit the word asthma control and rewrite “(f) having the level of asthma control at Global initiative For Asthma (GINA) step 3 and below” to “(f) is Global initiative For Asthma (GINA) step 3 and below”.

The word asthma control has been omitted and rewrote as suggested.

Factor extraction

Please elaborate and cite the decision to use the fixed number of factors method

The elaboration and citation for this decision has been added under Step 2: Factor extraction, retention and rotation section (page 12-13).

“In factor retention, many researchers use multiple criteria to decide on factor retention, which include parallel analysis, the Kaiser criterion (eigenvalue > 1), and the scree plot, with the latter two being the most commonly used (36). In this study, the factor retention was determined using parallel analysis, scree plot and fixed number of factors. Kaiser criterion method was not used as the original MAUQ has went through psychometric analysis which produced three factor solution (22). Thus, the domain of the items was preset to three as per original questionnaire before the factor analysis. Considering these two factors, and recommendation by the statistical experts, the fixed factor method was chosen.”

Result

Table 1: Final round content validity index (second stage) for item validation for ADU-Q Please clarify the number of experts who participated in the final round of the content validation

The final round of content validation involve 3 experts. The initial table was based on the first round of content validation, not the final round. This has been corrected in the manuscript (page 17-18).

Table 3 – Please substitute the word (GINA) “Stage” to “Step” in this table and throughout the manuscript

The word stage 3 has been changed to step 3 throughout the manuscript.

Discussion

Page 25, first paragraph: “The mean average usability score for ADU-Q was 5.58 (SD ±1.19)”. Please clarify where readers could find this in the results section. Please elaborate on the total score permissible for the ADU-Q and what scores mean (higher, lower, categories, if any)

This discussion part was removed as this study is assessing the validity of ADU-Q and not the usability of the asthma diary.

Page 25, first paragraph: “The mean average usability score for ADU-Q was 5.58 (SD ±1.19)”. Please clarify whether this reflects the usability of the ‘Buku Urus Kendiri Asma’ or something else

This discussion part was removed as this study is assessing the validity of ADU-Q and not the usability of the asthma diary.

Page 27, first paragraph: “In factor retention, several methods are used…”. Please consider moving this paragraph to a relevant subsection under Methods section

This has been moved to Statistical Analysis section under Step 2: Factor extraction, retention and rotation pages 12-13.

Limitation

“There are four limitations to this study.” Please check and make clear whether you have listed these four limitations

The limitation paragraph has been rechecked and amended.

Please clarify on whether or not the small number of expert panels in the content validation stage is a potential limitation

The first round of content validation comprised of recommended number of expert panel. The I-CVI result from first round was 1.0 but with some recommendation. In view of no major changes was required that affect the item itself, number of expert reduce to three. This per say is not a limitation as the I-CVI value achieved the cutoff point for good content validity which is 1.0.

Methods, Results and Discussions can be re-written more concisely. I recommend touching-up the manuscript for typos i.e. “dairy (page 3)”, grammatical errors and better sentence structure, for the readers’ benefits

Thank you for the suggestion. Some areas in the methods, results and discussion have been rechecked. The manuscript has been screened again for grammatical error and has been corrected.

REVIEWER 2

Comments Response

The abstract should elaborate more on the background of the study. Perhaps including the current situation on the control of asthma among the Malaysian population which further emphasize the urge to conduct this research. The content of abstract was not equally divided according to subsection. Most of the content are elaborating on the methods part. Perhaps the results and discussion parts should be refine in the abstract for the readers to get a better overview of the research.

The abstract has been added. Situation regarding asthma prevalence and control has been added in the abstract.

The methods and results section have been amended.

Introduction

In the first paragraph of the introduction, is it possible to add more details (RESONANCE study, no. of patients included with age range and other details such as comorbidities). The details need not be too long, it can be summarised within one or two sentences

More details regarding RESONANCE study has been added under the Introduction section, page 3.

“According to the RESONANCE study which included individuals age 12 years and above from a large populational databases in France, patients with uncontrolled asthma are two times more likely to be at risk of death compared to those with controlled asthma (6).”

In terms of healthcare burden, the cost to treat patients with uncontrolled asthma is higher than those with controlled asthma'. Is this sentence referring to the Malaysian population or other studies?. The following sentence on the costs of urgent care and maintenance care in Malaysia is also confusing as the costs were stated in USD. Perhaps the costs in both currencies can be stated

This sentence mainly referred to the cost in Malaysia which was part of the result of the cost of asthma in Asia-Pacific region study. The currency of USD has been converted into Malaysia ringgit. Both currencies were included in the sentence under Introduction section, page 3.

“In terms of healthcare burden in Malaysia, approximately RM 301 (USD 68) was spent per patient for urgent care compared to only RM 177 (USD 40) per patient for maintenance care (7).”

Paragraph 3 and 4 are disconnected. The use of asthma self-management booklet was developed in 2016 and there had been an updated version in 2023 indicating some precipitating issues/problems leading to the update of the booklet. This could be related to the need for usability assessment. Elaboration on this will make a better flow

The explanation regarding the booklet has been moved to ‘The study tool’ section. The problems related to the book has been included as well (page 6).

“This Asthma Self-Management Booklet was refined in 2023 after qualitative feedback from both physicians and asthma patient that have used the booklet. The refinement included improvement in the size of font and layout, asthma education content and graphics to make it more user friendly.”

Methods

The author may want to spell out FMS in figure 1 and use abbreviations in the following paragraphs

Figure 1 has been shifted to the end of method section and the footnotes for the abbreviation has been added (page 15)

Results

Put the reference range underneath each table for ease of reference (factor loading>0.6 indicates items validity)

The reference has been added under each table for ease of reference.

Discussion

There were several definitions given that could have been mentioned in the introduction instead to allow other important points to be included in the discussion e.g. elaboration on the results

The definition of several terms have been removed to relevant sections to allow more focus discussion on the result of the study.

Some parts of the discussion are redundant with the results for example "Test with Cronbach Alpha of >0.7 is considered reliable" and "overall ICC value was > 0.9 indicating good reproducibility and stability of the

score over time.

The relevant discussion sections have been amended to avoid redundancy.

To add discussion on the demographic of the participants

The discussion regarding the demographic of the participants has been added in the second paragraph (page 27-28)

Strength and Limitation

In the results part it was stated that during the data collection, a copy of the Asthma Self-Management Booklet and an explanation regarding the usage of the book were given to the patient. Would there be any bias imposed to the data collection process.

The explanation is mainly about how to fill in the asthma diary. All participants were given the same Google form link and the participant can remain anonymous when answering the form to avoid the researcher from influencing the response given by the participants.

Statistical Analysis

It was stated that all items in the questionnaire went through principal component analysis. Is this related to determining whether the factors are correlated before deciding to use promax rotations for analysis?. Is learnability strongly correlated with efficiency and satisfaction? etc.

The items went through principal component is to determine dimension reduction. This step is taken to compare the adapted questionnaire factor reduction and the factor produce by the original questionnaire. The promax rotation was used as according to literature, the elements of usability intercorrelated with each other.

Thank you very much for the responses.

---

## [Decision Letter · Decision Letter 1]

21 Nov 2025

Dear Dr. Nik Mohd Nasir,

Thank you for submitting your manuscript to PLOS ONE. After careful consideration, we feel that it has merit but does not fully meet PLOS ONE’s publication criteria as it currently stands. Therefore, we invite you to submit a revised version of the manuscript that addresses the points raised during the review process.

**ACADEMIC EDITOR:**

We look forward to receiving your revised manuscript.

Kind regards,

Mohd Ismail Ibrahim, MCom.Med

Academic Editor

PLOS ONE

Journal Requirements:

Reviewers' comments:

Reviewer's Responses to Questions

**Comments to the Author**

Reviewer #1: (No Response)

Reviewer #2: All comments have been addressed

2. Is the manuscript technically sound, and do the data support the conclusions?

Reviewer #1: Yes

Reviewer #2: Yes

3. Has the statistical analysis been performed appropriately and rigorously?

Reviewer #1: Yes

Reviewer #2: Yes

4. Have the authors made all data underlying the findings in their manuscript fully available?

Reviewer #1: Yes

Reviewer #2: Yes

5. Is the manuscript presented in an intelligible fashion and written in standard English?

Reviewer #1: No

Reviewer #2: Yes

Reviewer #1: Congratulations on producing a review of the manuscript. My comments serve to improve clarity and reader’s understanding.

Title:

Title clearly reflects the study scope and studied population, though slightly plain. If permitted, consider rephrasing the title. The use of a colon (:) in the title improves paper’s visibility in database searches.

Abstract:

Please rewrite the abstract guided by a structured format (Background -> Main Objective -> Methods -> Results -> Conclusion). An abstract should provide a concise overview of the study process whereas your abstract is a narrative and may appear too detailed. Recommend making more concise.

Introduction:

The background of study is sufficient to demonstrate to extent of problem, the role of asthma self-management and that a tool should be usable. However, please introduce the actual asthma diary itself as a foreshadowing to the aim of study. I suggest moving and rewriting the paragraph “The asthma diary used in this research was taken from The Asthma Self-Management Booklet…” from Methods (pg 6) to Introduction, in order to improve readers’ understanding of the study (which is to adapt and validate the M-MAUQ into the ADU-Q).

Methods:

If I understand correctly, the novelty of this research is the development of ADU-Q. However, the intention of the ADU-Q is first described only at the end of an extensive paragraph on M-MAUQ (pg 5). Please restructure the paragraph to begin with introducing the development of ADU-Q. In addition, please clarify whether all questions were adapted exclusively from the M-MAUQ. I suggest omitting the section “Study Tool” (pg 5) and moving contents to Phase I Adaptation and Content Validation (pg 6).

In Content Validation, “The experts were required to discuss and assess the relevance…” (pg. 7) , please clarify whether this is a form of Nominal Group Technique.

Based on the calculated I-CVI of the expert panel and their feedback, “the items that were deemed irrelevant or unclear were revised, modified or excluded as appropriate according to the expert panels’ suggestions” (pg. 8). Please clarify on how you decide whether which item gets omitted and which item gets iterated. Is there an objective measure (score/threshold level) as a guide or otherwise?

Reaching consensus is defined as “generally accepted opinion or decision among a group of people. In this case all experts agree….” (pg. 9). Please specify if there is an objective measurement of consensus e.g all items achieving a specific S-CVI or I-CVI value?

Please rephrase Figure 1 (pg. 15) concisely according to manuscript standard. Consider to alternately use concisely labeled pictograms to describe the study process instead.

Discussion

“The asthma diary has been part of various asthma guideline recommendation as a self-management tool for patients but a study by Tsang et al demonstrated that compliance to completing the asthma diary was low with only a quarter of participants still monitoring after six months. Thus, the lack of a suitable tool to assess the asthma diary usability has limited our ability to identify if this could be the potential reason for suboptimal compliance to improve on its usage….” (pg. 27) �This is a great study justification, and should be moved from Discussion to Introduction.

“However, to ensure that the participants remain homogenous, the inclusion criteria was set at GINA step 3 and excluded those with other underlying lung diseases” (pg. 28). Please correct to “GINA Step 3 and below”.

Please include that reliance of patient’s self-reporting is a limitation of this study however, as usability reflects user self-reflection, self-report is the most appropriate method for the study (pg 29)

Overall

Well done on producing valid and substantive research and manuscript. However, I strongly recommend a thorough language edit to polish this manuscript grammatically, improve sentence structure, flow and in alignment with scientific writing style. Thank you.

Reviewer #2: Dear Authors,

Thank you for addressing all previous comments. Here are some very minor additional comments.

Introduction part - In the first paragraph, it was stated that healthcare burden in Malaysia was approximately RM301 per patient. Can you clarify whether this is per month or per annum?

Methods- This part was clearly detailed out. Just one minor comment. Under sampling method, patient recruitment and data collection, can you elaborate on "clear verbal instruction" whether the instruction was given by a sole researcher/trained data collectors and whether the instruction is scripted or not.

Results- tables were self-explanatory. Table 3 under results (Gender), did not sum up to 100%. Male gender should be 21.9%. Table 5- Please correct the cronbach's alpha spelling.

Discussion- Please add year of publication for Tsang et al. (Para 1). I felt that the last paragraph under discussion was abruptly stopped. Conclude the discussion for a better flow.

That's all.

**Do you want your identity to be public for this peer review?** For information about this choice, including consent withdrawal, please see our Privacy Policy

Reviewer #1: No

Reviewer #2: **Yes:** Aina Yazrin Ali Nasiruddin

---

## [Author Response · Author response to Decision Letter 2]

4 Feb 2026

Reviewer’s comments for PONE-D-24-32929R1

Congratulations on producing a review of the manuscript. My comments serve to improve clarity and reader’s understanding.

We appreciate your constructive and valuable feedback. Please find our responses to your feedback below.

Title:

Title clearly reflects the study scope and studied population, though slightly plain. If permitted, consider rephrasing the title. The use of a colon (:) in the title improves paper’s visibility in database searches.

We have rephrased the title to: ‘Psychometric evaluation of the Asthma Diary Usability Questionnaire (ADU-Q): An adaptation and validation of a usability assessment tool among Adults with Asthma’

Abstract:

Please rewrite the abstract guided by a structured format (Background � Main Objective � Methods � Results � Conclusion). An abstract should provide a concise overview of the study process whereas your abstract is a narrative and may appear too detailed. Recommend making more concise.

We have rewritten the abstract as per recommendation summarizing concise overview of the study.

Introduction:

The background of study is sufficient to demonstrate to extent of problem, the role of asthma self-management and that a tool should be usable. However, please introduce the actual asthma diary itself as a foreshadowing to the aim of study. I suggest moving and rewriting the paragraph “The asthma diary used in this research was taken from The Asthma Self-Management Booklet…” from Methods (pg 6) to Introduction, in order to improve readers’ understanding of the study (which is to adapt and validate the M-MAUQ into the ADU-Q).

The paragraph in the Introduction has been rewritten as recommended as a foreshadowing to the aim of the study.

Methods:

If I understand correctly, the novelty of this research is the development of ADU-Q. However, the intention of the ADU-Q is first described only at the end of an extensive paragraph on M-MAUQ (pg 5). Please restructure the paragraph to begin with introducing the development of ADU-Q. In addition, please clarify whether all questions were adapted exclusively from the M-MAUQ. I suggest omitting the section “Study Tool” (pg 5) and moving contents to Phase I Adaptation and Content Validation (pg 6).

Yes, we have restructured the paragraph to introduce the development of ADU-Q, which was an adaptation from M-MAUQ alone.

In Content Validation, “The experts were required to discuss and assess the relevance…” (pg. 7) , please clarify whether this is a form of Nominal Group Technique.

Yes this is a type of consensus method which is a form of nominal group technique.

Based on the calculated I-CVI of the expert panel and their feedback, “the items that were deemed irrelevant or unclear were revised, modified or excluded as appropriate according to the expert panels’ suggestions” (pg. 8). Please clarify on how you decide whether which item gets omitted and which item gets iterated. Is there an objective measure (score/threshold level) as a guide or otherwise?

Yes, in the instructions to the panel, the items which are deemed not relevant to asthma diary usability based on their expert opinion are to be given scores 1 or 2 and are to be omitted from the ADU-Q. Items that are relevant but require some modifications are to be given the score of 3.

Reaching consensus is defined as “generally accepted opinion or decision among a group of people. In this case all experts agree….” (pg. 9). Please specify if there is an objective measurement of consensus e.g all items achieving a specific S-CVI or I-CVI value?

Yes, the threshold for achieving consensus for all items are set at I-CVI of at least 0.8.

Please rephrase Figure 1 (pg. 15) concisely according to manuscript standard (poor scientific writing). Consider to alternately use concisely labeled pictograms to describe the study process instead.

We have rephrased Figure 1 to make it more concise according to manuscript standard.

Discussion

“The asthma diary has been part of various asthma guideline recommendation as a self-management tool for patients but a study by Tsang et al demonstrated that compliance to completing the asthma diary was low with only a quarter of participants still monitoring after six months. Thus, the lack of a suitable tool to assess the asthma diary usability has limited our ability to identify if this could be the potential reason for suboptimal compliance to improve on its usage….” (pg. 27) �This is a great study justification, and should be moved from Discussion to Introduction.

We have moved this to Introduction.

“However, to ensure that the participants remain homogenous, the inclusion criteria was set at GINA step 3 and excluded those with other underlying lung diseases” (pg. 28). Please correct to “GINA Step 3 and below”.

We have made this correction.

Please include that reliance of patient’s self-reporting is a limitation of this study however, as usability reflects user self-reflection, self-report is the most appropriate method for the study (pg 29)

Thank you for the input, we have included this in the limitation of the study.

Overall

Well done on producing valid and substantive research and manuscript. However, I strongly recommend a thorough language edit to polish this manuscript grammatically, improve sentence structure, flow and in alignment with scientific writing style. Thank you.

We have improved on the language of the manuscript in alignment with scientific writing style.

Thank you for your feedback.

---

## [Editor Report · Decision Letter 2]

9 Feb 2026

Psychometric evaluation of the Asthma Diary Usability Questionnaire (ADU-Q): An adaptation and validation of a usability assessment tool among Adults with Asthma

PONE-D-24-32929R2

Dear Dr. Nik Mohd Nasir,

We’re pleased to inform you that your manuscript has been judged scientifically suitable for publication and will be formally accepted for publication once it meets all outstanding technical requirements.

Kind regards,

Mohd Ismail Ibrahim, MCom.Med

Academic Editor

PLOS One
---

## [Editor Report · Acceptance letter]

PONE-D-24-32929R2

PLOS One

Dear Dr. Nik Mohd Nasir,

I'm pleased to inform you that your manuscript has been deemed suitable for publication in PLOS One. Congratulations! Your manuscript is now being handed over to our production team.

Kind regards,

on behalf of

Dr. Mohd Ismail Ibrahim

Academic Editor

PLOS One